# Approximate Learning of High Dimensional Bayesian Network Structures via Pruning of Candidate Parent Sets

**DOI:** 10.3390/e22101142

**Published:** 2020-10-10

**Authors:** Zhigao Guo, Anthony C. Constantinou

**Affiliations:** 1Bayesian Artificial Intelligence Research Lab, School of Electronic Engineering and Computer Science, Queen Mary University of London, London E1 4NS, UK; 2The Alan Turing Institute, British Library, 96 Euston Road, London NW1 2DB, UK

**Keywords:** structure learning, probabilistic graphical models, pruning

## Abstract

Score-based algorithms that learn Bayesian Network (BN) structures provide solutions ranging from different levels of approximate learning to exact learning. Approximate solutions exist because exact learning is generally not applicable to networks of moderate or higher complexity. In general, approximate solutions tend to sacrifice accuracy for speed, where the aim is to minimise the loss in accuracy and maximise the gain in speed. While some approximate algorithms are optimised to handle thousands of variables, these algorithms may still be unable to learn such high dimensional structures. Some of the most efficient score-based algorithms cast the structure learning problem as a combinatorial optimisation of candidate parent sets. This paper explores a strategy towards pruning the size of candidate parent sets, and which could form part of existing score-based algorithms as an additional pruning phase aimed at high dimensionality problems. The results illustrate how different levels of pruning affect the learning speed relative to the loss in accuracy in terms of model fitting, and show that aggressive pruning may be required to produce approximate solutions for high complexity problems.

## 1. Introduction

A Bayesian Network (BN) [1] is a probabilistic graphical model represented by a Directed Acyclic Graph (DAG). The structure of a BN captures the relationships between nodes, whereas the conditional parameters capture the type and magnitude of those relationships. A BN differs from other graphical models, such as neural networks, in that it offers a transparent representation of a problem where the relationships between variables (i.e., arcs) represent conditional or causal relationships. Moreover, the uncertain conditional distributions in BNs can be used for both predictive and diagnostic (i.e., inverse) inference, providing the potential for a higher level of artificial intelligence. For example, knowledge-based BNs are often assumed to represent causal or influential networks and enable decision makers to reason about intervention [2]. On the other hand, structure of BNs, and especially those based on search-and-score solutions on which this paper focuses, are generally assumed to represent networks with conditional—rather than causal—relationships, although the class of constraint-based learning (which we cover below) is often used to discover relationships that could, under various assumptions, be interpreted causally [3].

Formally, a BN model represents a factorisation of the joint distribution of random variables $X=(X_{1},X_{2},\cdots ,X_{n})$. Each BN has two elements, structure *G* and parameters θ. Constructing a BN involves both structure learning and parameter learning, and both learning approaches may involve combination of data with knowledge [4,5,6,7]. Given observational data *D*, a complete BN {*G*, θ} can be learnt by maximising the likelihood:(1)P(G,θ|D)=P(G|D)P(θ|G,D).
and the parameters of the network can be learnt by maximising
(2)$$P\left(\theta |G,D\right)=\prod_{i=1}^{n}P\left(\theta_{i}|\Pi_{i},D\right)$$
where $\Pi_{i}$ denotes the parents of node $X_{i}$ indicated by structure *G*. Depending on the prior assumption, the parameter learning process of each node can be solved using Maximum Likelihood estimation given data *D*, or Maximum A Posteriori estimation given data *D* and a subjective prior.

On the other hand, the BN structure learning (BNSL) problem represents a more challenging task in that it cannot be solved by simply maximising the fitting of the local networks to the data. The structure learning process must take into consideration the complexity of the model in order to avoid overfitting. In fact, the problem of BNSL is NP-hard, which means that it is generally not possible to perform exhaustive search in the search space of possible graphs, and this is because the number of possible structures grows super-exponentially with the number of nodes *n* [8]:(3)$$f\left(n\right)=\sum_{i=1}^{n}{\left(-1\right)}^{i+1}\frac{n!}{(n-i)!n!}2^{i(n-i)}f\left(n-1\right)$$

Algorithms that learn BN structures are typically classified into three categories: (a) score-based learning that searches over the space of possible graphs and returns the graph that maximises a scoring function, (b) constraint-based learning that prunes and orientates edges using conditional independence tests, and (c) hybrid learning that combines the above two strategies. In this paper, we focus on the problem of score-based learning.

A score-based algorithm is generally composed of two parts: (a) a search strategy that transverses the search space, and (b) a scoring function that evaluates a given graph with respect to the observed data. Well-known search strategies in the field of BNSL include the greedy hill-climbing search, tabu search, simulated annealing, genetic algorithms, dynamic programming, A* algorithm, and branch-and-bound strategies. The objective functions are typically based on either the Bayesian score or other model selection scores. Bayesian scores evaluate the posterior probability of the candidate structures and include variations of the Bayesian Dirichlet score, such as the Bayesian Dirichlet with equivalence and uniform priors (BDeu) [9,10], and the Bayesian Dirichlet sparse (BDs) [11]. Well-established scores for model selection include the Akaike Information Criterion (AIC) [12] and the Bayesian Informatic Criterion (BIC), often also referred to as the Minimum Description Length (MDL) [13]. Other less popular model selection scores include the Mutual Information Test (MIT) [14], the factorized Normalized Maximum Likelihood (fNML) [15], and the qutotient Normalized Maximum Likelihood (qNML) [16].

Score-based approaches further operate in two different ways. The first approach involves scoring a graph only when the graph is visited by the search method, which typically involves exploring neighbouring graphs and following the search path that maximises the fitting score via arc reversals, additions and removals. The second approach involves generating scores for local networks (i.e., a node and its parents) in advance, and searching over combinations of local networks given the pre-generated scores, thereby formulating a combinatorial optimisation problem. The algorithms that fall in the former category are generally based on efficient heuristics such as hill-climbing, but tend to stuck in local optimum solutions, thereby offering an approximate solution to the problem of BNSL. While algorithms of the latter category are also generally approximate, they can be more easily adjusted to offer exact learning solutions that guarantee to return a graph with score not lower than the global maximum score. This paper focuses on this latter subcategory of score-based learning.

Algorithms such as the Integer Linear Programming (ILP) [17,18] explore local networks in the form of the Candidate Parent Sets (CPSs), usually up to a bounded maximum in-degree, and offer an exact solution. Other exact learning algorithms which cast the BNSL problem as a combinatorial optimisation problem include the Dynamic Programming (DP) [19,20], the A* algorithm [21] and the Branch-and-Bound (B&B) [22,23]. However, exact learning is generally restricted to problems of low complexity. Evidently, the efficiency of these algorithms is determined by the number of CPSs. For example, the ILP algorithm is restricted to CPSs of size up to one million. However, order-based algorithms such as OBS [24], ASOBS [25,26] and MINOBS [27] explore in the node ordering space, in which the number of structures consistent with the orderings that consist of *n* nodes is [28]
(4)$$f\left(n\right)=\prod_{i=1}^{n}2^{n-1}=2^{n(n-i)/2}.$$

Compared with Equation (Equation 3), the number of structures is substantially reduced. For example, for 10 nodes, there are $3.5\times 10^{13}$ different structures instead of $4.2\times 10^{18}$ computed by Equation (Equation 3). Therefore, order-based algorithms are able to search for approximate solutions in problems that involve hundreds of variables.

In this paper we focus on score-based algorithms that learn BN graphs via local learning of CPSs. Specifically, we investigate the relationship between different levels of pruning on CPSs and the loss in accuracy in terms of the BDeu score. The remainder of this paper is organised as follows: Section 2 provides the problem statement and methodology, Section 3 provides the results, and we provide our concluding remarks and directions for future research in Section 4.

## 2. Problem Statement and Methodology

Different pruning rules have been proposed to improve the scalability and the efficiency of score-based algorithms that operate on CPSs. Pruning approaches in this context generally aim to reduce the number of CPSs [22,29,30,31,32]. The efficacy of a pruning strategy can be significant in the reduction of computational complexity, and this depends on the number of the variables, the level of maximum in-degree and the number of observations in the data. For example, Table 1 presents a sample of the CPSs of node “0” in the Audio-train data set (https://github.com/arranger1044/awesome-spn#dataset), which consists of 100 variables and 15,000 observations. If we assume maximum-in-degree of 1, with no pruning, this produces $100\cdot (C_{99}^{0}+C_{99}^{1})=10,000$ CPSs, whereas for maximum-in-degree of 2 increases to $100\cdot (C_{99}^{0}+C_{99}^{1}+C_{99}^{2})=495,100$ CPSs, and for maximum in-degree of 3 increases to $100\cdot (C_{99}^{0}+C_{99}^{1}+C_{99}^{2}+C_{99}^{3})=16,180,000$ CPSs.

Pruning rules can be used to discard CPSs that are impossible to exist in optimal structures. Based on the observation that a parent set cannot be optimal if its subsets have higher scores [24], the CPSs that are left after applying these pruning rules are called “legal CPSs”. Table 2 presents an example where, in a network with four nodes {1,2,3,4}, part of the CPSs (those in bold) are pruned with the remaining CPSs representing the so-called legal CPSs. Figure 1 also illustrates all the possible CPSs of node 1, given a maximum in-degree of 3. As shown in Table 2, the CPSs, {2,3}, {2,4} and {2,3,4} are pruned and do not form part of the legal CPSs of node 1 because a) the score of CPS {2,3} is lower than its subset {3}, the score of CPS {2,4} is lower than its subset {4}, and the score of CPS {2,3,4} is lower than its subset {3,4}. In total, six CPSs are pruned out of the 32 possible CPSs. Figure 2 presents the optimal structure and shows that none of the pruned CPSs (or that only the legal CPSs) are part of the optimal structure.

We use the GOBNILP software (https://www.cs.york.ac.uk/aig/sw/gobnilp/) to obtain the legal CPSs. For example, the legal CPSs for the Audio-train data set, under maximum in-degree of 3, is 7,343,077 which represents a 45.4% of the total number of all possible CPSs. Table 3 presents the number and rates of legal CPSs, in relation to the number of all possible CPSs, over different sample sizes of the Audio-train data set and varied maximum in-degrees. The results show that the level of pruning decreases with sample size and increases with maximum in-degree. This is because a higher sample size leads to the detection of more dependencies whereas a higher maximum in-degree produces a higher number of possible dependencies that need to be explored. Still, this level of pruning is insufficient for high dimensionality problems. In general, the combinatorial optimisation problem of CPSs remains unsolvable for data sets that are large in terms of the number of the variables, with higher levels of maximum in-degree and data sample size further increasing complexity.

In this paper, we investigate the effect of different levels of pruning on legal CPSs. The effect is investigated both in terms of the gain in speed and the loss in accuracy, where the loss in accuracy is measured as a discrepancy Δ between the fitting scores of two learnt graphs defined as
(5)$$\Delta =\left(S^{*}-S\right)/S^{*}$$
where *S* denotes the BDeu score of a graph generated frompruning, and $S^{*}$ denotes the BDeu score of the baseline graph generated without pruning. This approach is based on the B&B algorithm proposed by Cassio de Campos [22], where the construction of the graph iterates over possible CPSs and starts from the most likely CPS per node. Specifically, we explore the CPSs expressed in the format shown in Table 1, where legal CPSs for each node are sorted in a descending order as determined by the local BDeu score. Different levels of pruning are explored by pruning different percentages of legal CPSs for each node, starting from the bottom-ranked CPSs of each node in terms of BDeu score. This means that the search in the space of possible DAGs starts from the most promising parent sets of each node. A valid DAG is ensured by skipping CPSs that lead to cycles. A valid DAG is achieved by ensuring that the learnt DAG incorporates at least one root node, which is a node having no parents [33]. This means that the exploration of CPSs must also include ‘empty’ CPSs (i.e., no parent), as shown in Table 1.

Three different levels of complexity are investigated, where the pruned result is compared to the unpruned result. The unpruned result represents the exact result for moderate complexity problems, although we cannot guarantee this will be the case for higher complexity problems. We define the three different levels of complexity as follows:(a)Moderate complexity, which assumes less than 1 million legal CPSs per network;(b)High complexity, which assumes more than 1 million and less than 10 million legal CPSs per network;(c)Very high complexity, which assumes more than 10 million legal CPSs per network.

We generate BDeu scores using the GOBNILP software and search for the optimal CPSs using different algorithms. Experiments on networks of moderate and high complexity were carried on an Intel Core i7-8750H CPU at 2.2 GHz with 16 GB of RAM, where each optimisation is assigned a maximum 9.2 GB of memory. Experiments on networks of very high complexity were carried on an Intel Core i7-8700 CPU at 3.2 GHz with 32 GB of RAM, where each optimisation is assigned a maximum 25 GB of memory. All experiments are restricted to 24 h of structure learning runtime.

## 3. Results

The experiments presented in this section are based on BNs and relevant data that are available on the GOBNILP website; link (https://www.cs.york.ac.uk/aig/sw/gobnilp/#benchmarks) for the networks used in Section 3.1, and link (https://www.cs.york.ac.uk/aig/sw/gobnilp/ for the networks used in Section 3.1 and Section 3.3.)

### 3.1. Pruning Legal CPSs of Moderate Complexity

We start the investigation by focusing on BNSL problems of moderate complexity, which we restrict to networks with up to 1 million legal CPSs. Table 4 lists the six networks with their corresponding number of legal CPSs for each case study and sample size combination, assuming a maximum in-degree of 3.

Table 5 and Table 6 present the loss in accuracy from different levels of CPSs pruning. The effect is measured as a discrepancy Δ defined in Section 2. For example, in Table 5, 90% pruning of the CPSs on Asia with sample size 100 leads to a graph that deviates 6.7‰, or 0.67%, in terms of BDeu score from the unpruned baseline graph. Note that 90% pruning of CPSs implies that, for each node, only the 10% most relevant CPSs are considered in searching for the optimal graph.

Overall, the results suggest that higher sample sizes encourage more aggressive pruning. This is reasonable because a higher sample size implies that the ordering of legal CPSs is more accurate and hence, the pruning also becomes more accurate in terms of pruning the least relevant CPSs. The results show that, in most cases, the loss in accuracy increases faster when pruning exceeds the level of 30%. However, the results from the Hailfinder and Carpo networks suggest that even minor levels of pruning can have a negative impact on the BDeu score, however small this impact may be. All the moderate complexity experiments completed search within four minutes, except the case of Carpo-10000 which took approximately twelve minutes to complete. From this, we can conclude that unless the intention is to save seconds or minutes of structure learning runtime, pruning of legal CPSs is less desirable in problems of moderate complexity.

### 3.2. Pruning Legal CPSs of High Complexity

This subsection reports the results from pruning based on high complexity case studies that involve problems where the number of legal CPSs ranges between 1 million and 10 million. For this scenario, we used the real-world data sets called Audio-train which consists of 100 variables and 15,000 observations, and Kosarek-test which consists of 190 variables and 6675 observations. We used GOBNILP to generate the BDeu scores for CPSs. This process took 90 and 1398 seconds to complete respectively, with GOBNILP returning 7,343,077 and 5,748,931 legal CPSs, respectively.

Because GOBNILP’s ILP algorithm is restricted to CPSs of size less than 1 million, we replaced ILP with the approximate algorithm called MINOBS (https://github.com/kkourin/mobs). The change from exact to approximate learning was inevitable since exact solutions are only applicable to problems of relatively low complexity. In fact, the results show that even an approximate algorithm such as MINOBS fails to complete search within the 24-hour runtime limit in the absence of pruning. At 24 h of runtime, we stop the search and obtain the highest scoring graph discovered up to that point.

Table 7 presents the results from this set of experiments. The results suggest that problems of high complexity may benefit considerably from pruning compared to problems of moderate complexity. In fact, the results show that it may be safe to perform aggressive pruning on legal CPSs without, or with limited, loss in accuracy, in exchange for a considerable reduction in runtime. For example, 90% pruning on the CPSs of the Audio-train data set is found to reduce runtime needed to first discover the highest scoring graph by approximately 21 folds without, or in exchange for a trivial, reduction in the BDeu score. Note that while the time needed to first discover the highest scoring graph is generally expected to decrease with higher levels of pruning, Table 7 shows that this is generally, although not always, the case.

The increased benefit from pruning, observed in the case of the Audio-train and Kosarek-test data sets, relative to the data sets of moderate complexity investigated in Section 3.1, can be explained by the higher number of CPSs in the network. This is because when working with higher numbers of CPSs and a low bounded maximum in-degree (in this case, 3), even 90% pruning of legal CPSs makes it likely that the top three most relevant variables (out of hundreds of variables) will not be part of those pruned.

### 3.3. Pruning Legal CPSs of Very High Complexity

Lastly, we investigate the effect of pruning in case studies that incorporate more than 10 million legal CPSs. For this purpose, we used the EachMovie-train and Reuters-52-train data sets taken from the same repository. EachMovie-train consists of 500 variables and 4524 observations, whereas Reuters-52-train consists of 889 variables and 6532 observations. As with the high complexity cases, we perform the experiments using the MINOBS algorithm. The GOBNILP software generated a total of 21,985,307 and 37,479,789 legal CPSs, in 134 and 616 seconds respectively. However, in these experiments we had to reduce the maximum in-degree from 3 to 2. This was necessary to avoid running out of memory.

The results in Table 8 suggest that we can derive conclusions that are similar to those derived for the high complexity experiments in Section 3.2. Specifically, the pruning strategy appears to have a minor impact on accuracy of very high complexity network scores, in exchange for potentially large reductions in runtime. Note that while higher levels of pruning always reduce the time required to find the highest scoring graphs from those explored within the 24h runtime limit in the case of EachMovie-train, the effectiveness of pruning is not as consistent in the case of Reuters-52-train. This suggests that the pruning effectiveness also depends on the data set and may not be solely due to randomness as discussed in Section 3.2.

## 4. Conclusions

This study investigated the effectiveness of different levels of CPS pruning across problems of varied complexity. The results suggest that it is generally not beneficial to perform pruning of legal CPSs on problems of moderate or lower complexity. This is because the risk of pruning relevant CPSs increases in low complexity case studies that tend to incorporate a lower number of variables, in exchange for relatively minor improvements in speed. On the other hand, the results from problems of higher complexity show potential for major benefit from this type of pruning. This is because these problems tend to incorporate hundreds or thousands of variables, and such a high number of variables makes it easier to determine and prune irrelevant parent-sets, thereby minorly impacting accuracy in exchange for considerable gains in speed. Importantly, problems of very high complexity are often unsolvable and could benefit enormously from any form of effective pruning. The pruning strategy investigated in this paper applies to any type of score-based learning, including the traditional greedy hill-climbing heuristics where pruned legal CPSs could be used to restrict the path of arc additions.

Future work can be extended in various directions. Firstly, more experiments are needed to derive stronger conclusions about the effect of this type of pruning across different algorithms and hyperparameter settings (e.g., over different bounded maximum in-degree). Other research directions include investigating this type of pruning on ordered-based algorithms, where such a pruning strategy could be used to restrict the search space of ordered-based graphs. Lastly, other studies have shown that maximising an objective function does not necessarily imply a more accurate causal graph, especially when the data incorporate noise [34]. This diminishes the importance of exact learning and invites future work where the effect of pruning is judged in terms of graphical structure, in addition to its impact on a fitting score.

## Figures and Tables

**Figure 1 entropy-22-01142-f001:**
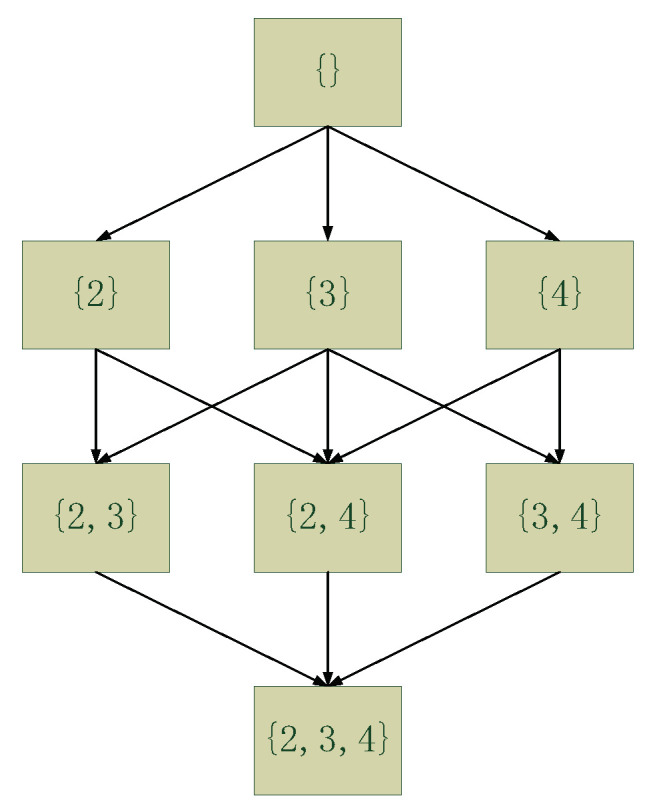
All possible CPSs of node “1” (not shown in the diagram) under the assumption the maximum in-degree is 3.

**Figure 2 entropy-22-01142-f002:**
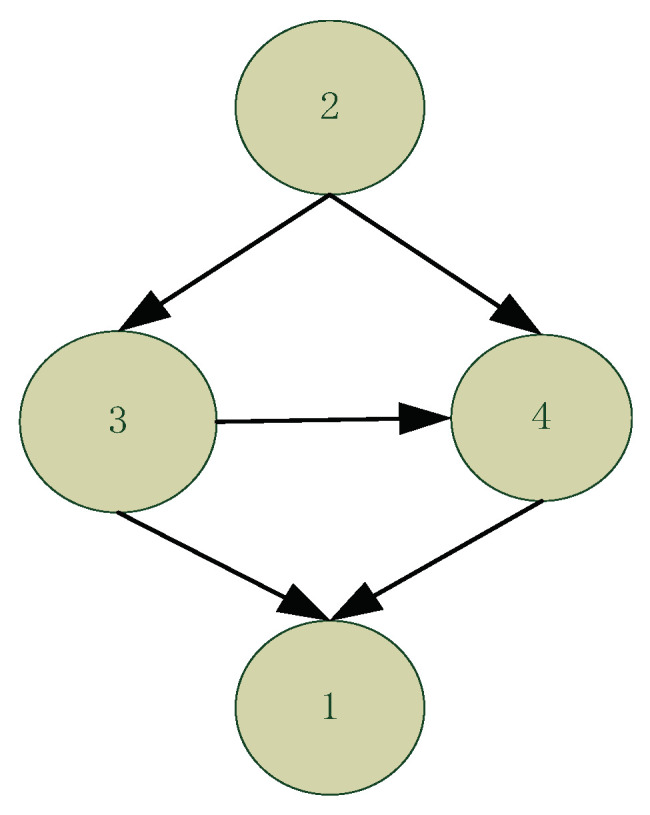
The optimal structure learnt from the CPSs presented in Table 2.

**Table 1 entropy-22-01142-t001:** Sample CPSs of node “0”, ordered by BDeu score with max in-degree 3. The example is based on Audio-train dataset which incorporates 100 variables.

Child Node	Local BDeu Score	CPS Size	CPS
0	−5149.19	3	{9, 85, 95}
0	−5150.47	3	{9, 94, 95}
0	−5174.53	3	{85,94, 95}
0	−5207.08	3	{80,85, 95}
0	−5208.28	3	{9, 80, 95}
…	…	…	…
0	−6886.30	2	{48,67}
0	−6886.74	1	{67}
0	−5174.53	1	{81}
0	−5174.53	1	{75}
0	−6889.11	0	{}

**Table 2 entropy-22-01142-t002:** An example BN with four nodes and the legal CPSs that remain after pruning the CPSs that are impossible to exist (highlighted in bold), as determined by the BDeu score. The example assumes four nodes, a maximum in-degree (ID) of 3, and a sample size of 5000.

Node	ID = 0	ID = 1	ID = 1	ID = 1	ID = 2	ID = 2	ID = 2	ID = 3
1	{}	{2}	{3}	{4}	{2,3}	{2,4}	{3,4}	{2,3,4}
	−2288.7	−2274.6	−2196.2	−2240.7	**−2252.8**	**−2256.1**	−2171.3	**−2173.5**
2	{}	{1}	{3}	{4}	{1,3}	{1,4}	{3,4}	{1,3,4}
	−2003.7	−1989.6	−1900.7	−1915.1	**−1903.8**	**−1918.3**	−1849.2	**−1851.4**
3	{}	{1}	{2}	{4}	{1,2}	{1,4}	{2,4}	{1,2,4}
	−2891.5	−2799.0	−2788.5	−2811.3	−2714.5	−2741.9	−2745.5	−2692.6
4	{}	{1}	{2}	{3}	{1,2}	{1,3}	{2,3}	{1,2,3}
	−1951.6	−1903.6	−1862.9	−1871.4	−1829.5	−1846.5	−1819.9	−1807.6

**Table 3 entropy-22-01142-t003:** The number and rates of legal CPSs in relation to the all possible CPSs for subsets of Audio-train data over varying samples sizes and maximum in-degrees.

Maximum In-Degree	Number of All Possible CPSs	Sample Size				
		3000	6000	9000	12,000	15,000
1	10,000	8398	8926	9163	9320	9394
		84.0%	89.3%	91.6%	93.2%	93.9%
2	495,100	228,197	306,263	349,587	374,007	388,621
		46.1%	61.9%	70.6%	75.5%	78.5%
3	16,180,000	1,200,429	3,260,399	5,130,502	6,405,394	7,343,077
		7.42%	20.2%	31.7%	39.6%	45.4%

**Table 4 entropy-22-01142-t004:** Moderate complexity case studies (nodes∣max in-degree in true networks), depicting the total number of legal CPSs per network, as well as the average number of CPSs per node in that network, for network and sample size combination. The number of legal CPSs assume a maximum in-degree of 3.

	Sample Size	Asia (8∣2)	Insurance (27∣3)	Water (32∣5)	Alarm (37∣4)	Hailfinder (56∣4)	Carpo (61∣5)
CPSs (graph)	100	41	279	482	907	244	5068
	1000	107	774	573	1928	761	3827
	10,000	161	3652	961	6473	3768	16,391
CPSs (per node)	100	5.13	10.33	15.06	24.51	4.36	84.47
	1000	13.38	28.67	17.91	52.11	13.59	63.78
	10,000	20.12	135.26	30.03	174.95	67.29	273.18

**Table 5 entropy-22-01142-t005:** Loss in accuracy for different levels of pruning, as a discrepancy Δ in BDeu score from the unpruned score, based on the three different sample sizes for case studies Asia, Insurance and Water.

Pruning	Asia (100)	Asia (1000)	Asia (10,000)	Insurance (100)	Insurance (1000)	Insurance (10,000)	Water (100)	Water (1000)	Water (10,000)
90%	−6.70‰	−1.26‰	−1.33‰	−30.74‰	−62.92‰	−35.26‰	−11.84‰	−28.11‰	−15.50‰
80%	−6.70‰	−1.26‰	−1.06‰	−30.74‰	−37.77‰	−7.99‰	−11.15‰	−19.37‰	−8.12‰
70%	−6.70‰	−1.26‰	−1.06‰	−10.50‰	−13.80‰	−7.13‰	−8.21‰	−2.99‰	−0.68‰
60%	−6.70‰	−1.26‰	−0.72‰	−8.32‰	−6.73‰	−5.32‰	−6.70‰	−2.81‰	−0.44‰
50%	−6.68‰	−1.26‰	−0.72‰	−7.94‰	−4.14‰	−2.83‰	−1.24‰	−1.02‰	−0.27‰
40%	−0.04‰	−1.26‰	−0.72‰	−2.33‰	−1.28‰	−2.07‰	−0.64‰	**0‰**	−0.18‰
30%	**0‰**	−0.9‰	−0.25‰	−2.23‰	**0‰**	−1.22‰	−0.32‰	**0‰**	−0.02‰
20%	**0‰**	**0‰**	**0‰**	**0‰**	**0‰**	−0.25‰	−0.32‰	**0‰**	**0‰**
10%	**0‰**	**0‰**	**0‰**	**0‰**	**0‰**	**0‰**	**0‰**	**0‰**	**0‰**
0%	**0‰**	**0‰**	**0‰**	**0‰**	**0‰**	**0‰**	**0‰**	**0‰**	**0‰**

**Table 6 entropy-22-01142-t006:** Loss in accuracy for different levels of pruning, as a discrepancy Δ in BDeu score from the unpruned score, based on the three different sample sizes for case studies Alarm, Hailfinder and Carpo.

Pruning	Alarm (100)	Alarm (1000)	Alarm (10,000)	Hailfinder (100)	Hailfinder (1000)	Hailfinder (10,000)	Carpo (100)	Carpo (1000)	Carpo (10,000)
90%	−78.39‰	−46.86‰	−23.13‰	−34.15‰	−8.21‰	−8.82‰	−7.88‰	−3.84‰	−2.90‰
80%	−30.04‰	−38.71‰	−14.44‰	−25.02‰	−4.17‰	−6.05‰	−5.29‰	−3.13‰	−1.99‰
70%	−18.87‰	−22.93‰	−3.88‰	−10.03‰	−4.17‰	−4.23‰	−4.33‰	−2.02‰	−1.94‰
60%	−13.55‰	−14.33‰	−1.99‰	−2.23‰	−2.16‰	−1.38‰	−4.33‰	−1.78‰	−1.85‰
50%	−4.27‰	−5.23‰	−1.79‰	−1.57‰	−1.60‰	−0.57‰	−3.97‰	−1.73‰	−1.10‰
40%	−3.69‰	−1.82‰	−0.20‰	−1.57‰	−1.03‰	−0.57‰	−2.33‰	−1.54‰	−1.06‰
30%	−1.06‰	−0.30‰	**0‰**	−1.27‰	−0.20‰	−0.06‰	−1.51‰	−1.17‰	−0.93‰
20%	−1.06‰	−0.30‰	**0‰**	−0.07‰	−0.19‰	−0.06‰	−1.01‰	−0.25‰	−0.35‰
10%	**0‰**	−0.15‰	**0‰**	**0‰**	−0.19‰	−0.06‰	−1.01‰	−0.18‰	−0.02‰
0%	**0‰**	**0‰**	**0‰**	**0‰**	**0‰**	**0‰**	**0‰**	**0‰**	**0‰**

**Table 7 entropy-22-01142-t007:** Loss in accuracy for different levels of pruning, as a discrepancy Δ in BDeu score from the unpruned score, based on the three different sample sizes for case studies Audio-train and Kosarek-test. Time (secs) represents the time needed by the MINOBS algorithm to first discover the highest scoring graph within the 24 h of search.

	Audio-Train				Kosarek-Test			
**Pruning**	**CPSs**		Δ	**Time (secs)**	**CPSs**		Δ	**Time (secs)**
	**Graph**	**per Node**			**Graph**	**per Node**		
99%	73,535	735	−4.352‰	1473	58,249	307	−7.468‰	4260
95%	367,258	3673	−0.669‰	682	287,641	1514	−0.271‰	3265
90%	734,414	7344	−0.002‰	1035	575,096	3027	**0‰**	1803
80%	1,468,717	14,687	**0‰**	2952	1,149,980	6053	**0‰**	16,378
70%	2,203,033	22,030	**0‰**	3908	1,724,881	9078	**0‰**	16,010
60%	2,937,329	29,373	**0‰**	5344	2,299,767	12,104	**0‰**	20,637
50%	3,671,663	36,717	**0‰**	4334	2,874,708	15,130	**0‰**	9033
40%	4,405,948	44,059	**0‰**	4587	3,449,544	18,155	**0‰**	14,903
30%	5,140,257	51,403	**0‰**	10,028	4,024,450	21,181	**0‰**	9288
20%	5,874,560	58,746	**0‰**	10,442	4,599,334	24,207	**0‰**	29,603
10%	6,608,876	66,089	**0‰**	11,385	5,174,238	27,233	**0‰**	42,493
0%	7,343,077	73,431	**0‰**	21,643	5,748,931	30,258	**0‰**	82,758

**Table 8 entropy-22-01142-t008:** Loss in accuracy for different levels of pruning, as a discrepancy Δ in BDeu score from the unpruned score, based on the three different sample sizes for case studies EachMovie-train and Reuters-52. Time (secs) represents the time needed by the MINOBS algorithm to first discover the highest scoring graph within the 24 h of search.

	EachMovie-Train				Reuters-52-Train			
**Pruning**	**CPSs**		Δ	**Time (secs)**	**CPSs**		Δ	**Time (secs)**
	**Graph**	**per Node**			**Graph**	**per Node**		
99%	220,378	441	−0.671‰	1711	375,700	423	−1.269‰	3368
95%	1,099,782	2200	−0.158‰	6471	1,874,897	2109	−0.051‰	6430
90%	2,199,065	4398	**0‰**	9049	3,748,921	4217	**0‰**	10,002
80%	4,397,558	8795	**0‰**	15,273	7,496,843	8433	**0‰**	34,537
70%	6,596,133	13,192	**0‰**	23,133	11,244,877	12,649	**0‰**	41,554
60%	8,795,121	17,589	**0‰**	9195	14,992,798	16,865	**0‰**	12,925
50%	10,993,281	21,943	**0‰**	15,812	18,741,002	21,081	**0‰**	27,914
40%	13,191,681	26,383	**0‰**	37,244	22,488,769	25,297	**0‰**	17,276
30%	15,390,238	30,780	**0‰**	74,312	26,236,772	29,513	**0‰**	72,208
20%	17,588,746	35,107	**0‰**	24,576	29,984,724	33,729	**0‰**	16,969
10%	19,787,306	39,575	**0‰**	35,952	33,732,728	37,945	**0‰**	69,315
0%	21,985,307	43,971	**0‰**	82,758	37,479,789	42,159	**0‰**	48,704

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
