# Peer review of "Approximate Learning of High Dimensional Bayesian Network Structures via Pruning of Candidate Parent Sets"

_entropy, 2020, doi:10.3390/e22101142_

Round 1
Reviewer 1 Report
The paper presents an experimental study on the influence of search space restriction by using constrained candidate parent set when learning Bayesian networks.
The paper does not provide anything new from the methodological point of view, but presents an experimental study that could be interesting.
Reviewer 2 Report
The authors explore a strategy for learning the structure of Bayesian Networks. The paper focuses on the score based algorithms, searching over a combination of local networks.
I think that the paper is interesting since it gives a good overview of Bayesian Networks learning based on score algorithms, suggest some solutions to deal with datasets which incorporate hundreds or thousands of variables, and it is well written.
I have just a minor comment related to the Introduction. The authors put casualty as keywords and say that the relation can be interpreted causally. However, it is necessary to put several assumptions to interpret the links in a bayesian network as causal relations. I would be more careful.
Reviewer 3 Report
1. The paper needs minor corrections:
- line 28: where, IIi ---> where IIi
- line 86: depends on both the ---> depends on the
- line 108: where, S ---> where S
- line 108: pruning ---> pruning,
- line 117: has ---> having
- line 161: CPSs. ---> CPSs, respectively.
- line 172: are ---> is
- line 177: pruning ---> pruning,
2. At line 181, when they authors say "will make it", it is not clear to me what "it" refers to.
3. In the captions of figures 6 and 7, the authors simply write "algorithm". In my opinion, the name of the algorithm should be mentioned.
4. The paper contains no figures. I suggest the introduction of an example of Bayesian Network learnt from one data set.
5. Several data sets are analysed in the paper (Asia, Insurance, Water, etc.). Is it possible to cite the source of the data sets? (URL or bibliographic reference)
